# Neural network gradient-based learning of black-box function interfaces

**Alon Jacovi**[1,2][*]**, Guy Hadash**[1][*]**, Einat Kermany**[1][*]**, Boaz Carmeli**[1][*]**,**
**Ofer Lavi**[1]**, George Kour**[1]**, Jonathan Berant**[3,4]
[1] IBM Research, Israel
[2] Bar Ilan University, Israel
[3] Tel Aviv University, Israel
[4] Allen Institute for Artificial Intelligence
alonjacovi@gmail.com
{guyh,einatke,boazc}@il.ibm.com

## Abstract

Deep neural networks work well at approximating complicated functions when provided with data and trained by gradient descent methods. At the same time, there is a vast amount of existing functions that programmatically solve different tasks in a precise manner eliminating the need for training. In many cases, it is possible to decompose a task to a series of functions, of which for some we may prefer to use a neural network to learn the functionality, while for others the preferred method would be to use existing black-box functions. We propose a method for end-to-end training of a base neural network that integrates calls to existing black-box functions. We do so by approximating the black-box functionality with a differentiable neural network in a way that drives the base network to comply with the black-box function interface during the end-to-end optimization process. At inference time, we replace the differentiable estimator with its external black-box non-differentiable counterpart such that the base network output matches the input arguments of the black-box function. Using this "Estimate and Replace" paradigm, we train a neural network, end to end, to compute the input to black-box functionality while eliminating the need for intermediate labels. We show that by leveraging the existing precise black-box function during inference, the integrated model generalizes better than a fully differentiable model, and learns more efficiently compared to RL-based methods.

## 1 Introduction

End-to-end supervised learning with deep neural networks (DNNs) has taken the stage in the past few years, achieving state-of-the-art performance in multiple domains including computer vision (Szegedy et al., 2017), natural language processing (Sutskever et al., 2014; Jean et al., 2015), and speech recognition (Xiong et al., 2016). Many of the tasks addressed by DNNs can be naturally decomposed to a series of functions. In such cases, it might be advisable to learn neural network approximations for some of these functions and use precise existing functions for others. Examples of such tasks include Semantic Parsing and Question Answering. Since such a decomposition relies partly on precise functions, it may lead to a superior solution compared to an approximated one based solely on a learned neural model.

Decomposing a solution into trainable networks and existing functions requires matching the output of the networks to the input of the existing functions, and vice-versa. The input and output are defined by the existing functions' interface. We shall refer to these functions as *black-box functions* (bbf), focusing only on their interface. For example, consider the question: "Is 7.2 greater than 4.5?" Given that number comparison is a solved problem in symbolic computation, a natural solution would be to decompose the task into a two-step process of (i) converting the natural language to an executable program, and (ii) executing the program on an arithmetic module. While a DNN may be a good fit for

---

[*]Equal contribution

the first step, it would not be a good fit for the second step, as DNNs have been shown to generalize poorly to arithmetic or symbolic functionality (Fodor & Pylyshyn, 1988; He et al., 2016).

In this work, we propose a method for performing end-to-end training of a decomposed solution comprising of a neural network that calls black-box functions. Thus, this method benefits from both worlds. We empirically show that such a network generalizes better than an equivalent end-to-end network and that our training method is more efficient at learning than existing methods used for training a decomposed solution.

The main challenge in decomposing a task to a collection of neural network modules and existing functions is that effective neural network training using gradient-based techniques requires the entire computation trajectory to be differentiable. We outline three existing solutions for this task: (i) *End-to-End Training:* Although a task is naturally decomposable, it is possible to train a network to fit the symbolic functionality of the task to a differentiable learned function without decomposing it. Essentially, that means solving the problem end-to-end, foregoing the existing black-box function. (ii) *Using Intermediate Labels:* If we insist, however, on using the black-box function, it is possible to train a network to supply the desired input to the black-box function by providing intermediate labels for translating the task input to the appropriate black-box function inputs. However, intermediate labels are, most often, expensive to obtain and thus produce a bottleneck in gathering data. (iii) *Black-Box Optimization:* Finally, one may circumvent the need for intermediate labels or differentiable approximation with Reinforcement Learning (RL) or Genetic Algorithms (GA) that support black-box function integration during training. Still, these algorithms suffer from high learning variance and poor sample complexity.

We propose an alternative approach called *Estimate and Replace* that finds a differentiable function approximation, which we term *black-box estimator*, for estimating the black-box function. We use the black-box estimator as a proxy to the original black-box function during training, and by that allow the learnable parts of the model to be trained using gradient-based optimization. We compensate for not using any intermediate labels to direct the learnable parts by using the black-box function as an oracle for training the black-box estimator. During inference, we replace the black-box estimator with the original non-differentiable black-box function.

End-to-end training of a solution composed of trainable components and black-box functions poses several challenges we address in this work—coping with non-differentiable black-box functions, fitting the network to call these functions with the correct arguments, and doing so without any intermediate labels. Two more challenges are the lack of prior knowledge on the distribution of inputs to the black-box function, and the use of gradient-based methods when the function approximation is near perfect and gradients are extremely small.

This work is organized as follows: In Section 2, we formulate the problem of decomposing the task to include calls to a black-box function. Section 3 describes the network architecture and training procedures. In Section 4, we present experiments and comparison to Policy Gradient-based RL, and to fully neural models. We further discuss the potential and benefits of the modular nature of our approach in Section 6.

## 2  LEARNING BLACK-BOX FUNCTION INTERFACES WITHOUT INTERMEDIATE LABELS

In this work, we consider the problem of training a DNN model to interact with black-box functions to achieve a predefined task. Formally, given a labeled pair $(x, y)$, such that some target function $h^* : X \to Y$ satisfies $h^*(x) = y$, we assume that there exist:

$$h_i^{\mathrm{arg}} : X \to A_i \;\; ; \;\; i \leq n \tag{1}$$

$$h^{\mathrm{bbf}} : (A_1, ..., A_n) \to Y \tag{2}$$

Such that $h^*(x) = h^{\mathrm{bbf}}(h_1^{\mathrm{arg}}(x), ..., h_n^{\mathrm{arg}}(x))$, where $n$ is the number of arguments in the black-box input domain $A = (A_1, ..., A_n)$. The domains $\{A_i\}$ can be structures of discrete, continuous, and nested variables.

The problem then is to fit $h^*$ given a dataset $\{(x^j, y^j) \mid j \leq D\}$ and given an oracle access to $h^{\mathrm{bbf}}$. Then $h^{\mathrm{arg}} : X \to (A_1, ..., A_n)$ is an *argument extractor function*, which takes as input $x$ and outputs a

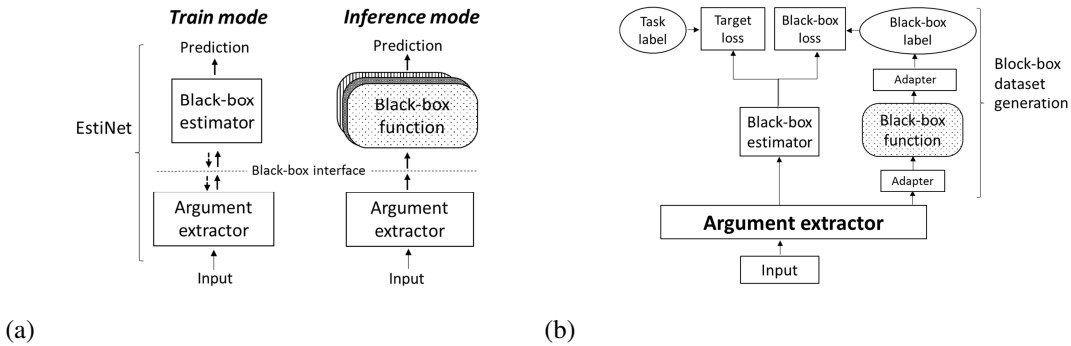

(a)                                     (b)

Figure 1: The *Estimate and Replace* approach. Figure (a) shows a schematic description of the two assessment modes. A rectangle represents a neural network. Solid and dashed arrows signify forward and backward passes of back-propagation. Figure (b) shows a more detailed schematic view of the model with regards to usage of the external function during training.

tuple of *arguments* $(h_1^{\mathrm{arg}}(x), ..., h_n^{\mathrm{arg}}(x)) = (a_1, ..., a_n)$, and $h^{\mathrm{bbf}}$ is a *black-box function*, which takes these arguments and outputs the final result. Importantly, we require no sample $(x, (a_1, ..., a_n), y)$ for which the intermediate black-box interface argument labels are available. We note that this formulation allows the use of multiple functions simultaneously, e.g., by defining an additional argument that specifies the "correct" function, or a set of arguments that specify ways to combine the functions' results.

## 3 THE ESTIMATE AND REPLACE APPROACH AND THE ESTINET MODEL

In this section we present the *Estimate and Replace* approach which aims to address the problem defined in Section 2. The approach enables training a DNN that interacts with non-differentiable black-box functions (bbf), as illustrated in Figure 1 (a). The complete model, termed *EstiNet*, is a composition of two modules—*argument extractor* and *black-box estimator*—which learn $h^{\mathrm{arg}}$ and $h^{\mathrm{bbf}}$ respectively. The black-box estimator sub-network serves as a differential estimator to the black-box function during an end-to-end gradient-based optimization. We encourage the estimator to directly fit the black-box functionality by using the black-box function as a label generator during training. At inference time, we replace the estimator with its black-box function counterpart, and let this hybrid solution solve the end-to-end task at hand in an accurate and efficient way. In this way, we eliminate the need for intermediate labels. We refer to running a forward-pass with the black-box estimator as *test mode* and running a forward-pass with the black-box function as *inference mode*. By leveraging the black-box function as in this mode, EstiNet shows better gerealization than an end-to-end neural network model. In addition, EstiNet suggests a modular architecture with the added benefits of module reuse and model interpretability.

**Adapters** EstiNet uses an adaptation function to adapt the argument extractor's output to the black-box function input, and to adapt black-box function's output to the appropriate final output label format (see Figure 1 (b)). For example, EstiNet uses such a function to convert soft classification distributions to hard selections, or to map classes of text token to concrete text.

### 3.1 TRAINING AN ESTINET MODEL

The modular nature of the EstiNet model presents a unique training challenge: EstiNet is a modular architecture where each of its two modules, namely the argument extractor and the black-box estimator is trained using its own input-label pair samples and loss function.

### 3.1.1 ESTINET'S LOSS FUNCTIONS AND DATASETS

We optimize EstiNet model parameters with two distinct loss functions—the *target loss* and the *black-box loss*. Specifically, we optimize the argument extractor's parameters with respect to the target loss using the task's dataset during end-to-end training. We optimize the black-box estimator's parameters with respect to the black-box loss while training it on the black-box dataset:

**The black-box dataset** We generate input-output pairs for the black-box dataset by sending an input sample to the black-box function and recording its output as the label. We experimented in generating input samples in two ways: (1) offline sampling—in which we sample from an a-priori black-box input distribution, or from a uniform distribution in absence of such; and (2) online sampling—in which we use the output of the argument extractor module during a forward pass as an input to the black-box function, using an adaptation function as needed for recording the output (see Figure 1 (b)).

### 3.1.2 TRAINING PROCEDURES

Having two independent datasets and loss functions suggest multiple training procedure options. In the next section we discuss the most prominent ones along with their advantages and disadvantages. We provide empirical evaluation of these procedures in Section 4.

**Offline Training** In offline training we first train the black-box estimator using offline sampling. We then fix its parameters and load the trained black-box estimator into the EstiNet model and train the argument extractor with the task's dataset and target loss function. A disadvantage of offline training is noisy training due to the *distribution difference* between the offline black-box a-priori dataset and the actual posterior inputs that the argument extractor computes given the task's dataset during training. That is, the distribution of the dataset with which we trained the black-box estimator is different than the distribution of input it receives during the target loss training.

**Online Training** In online training we aim to solve the distribution difference problem by jointly training the argument extractor and the black-box estimator using the target loss and black-box loss respectively. Specifically, we train the black-box estimator with the black-box dataset generated via online sampling during the training process.[1] Figure 1 (b) presents a schematic diagram of the online training procedure. We note that the online training procedure suffers from a cold start problem of the argument extractor. Initially, the argument extractor generates noisy input for the black-box function, which prevents it from generating meaningful labels for the black-box estimator.

**Hybrid Training** In hybrid training we aim to solve the cold start problem by first training the black-box estimator offline, but refraining from freezing its parameters. We load the estimator into the EstiNet model and continue to train it in parallel with the argument extractor as in online training.

### 3.1.3 REGULARIZING BLACK-BOX ESTIMATOR OVER-CONFIDENCE

In all of the above training procedures, we essentially replace the use of intermediate labels with the use of a black-box dataset for implicitly training the argument extractor via back-propagation. As a consequence, if the gradients of the black-box estimator are small, it will make it difficult for the argument extractor to learn. Furthermore, if the black-box estimator is a classifier, it tends to grow overly confident as it trains, assigning very high probabilities to specific answers and very low probabilities for the rest (Pereyra et al., 2017). Since these classification functions are implemented with a softmax layer, output values that are close to the function limits $(0, 1)$ result in extremely small gradients. Meaning that in the scenario where the estimator reaches local optima and is very confident, its gradient updates become small. Through the chain rule of back-propagation, this means that even if the argument extractor is not yet at local optima, its gradient updates become small as well, which complicates training.

---

[1] We note that this problem is reminiscent of, but different from, Multi-Task Learning, which involves training the same parameters using multiple loss functions. In our case, we train non-overlapping parameters using two losses: Let $L_{\text{target}}$ and $L_{\text{bbf}}$ be the two respective losses, and $\theta_{\text{arg}}$ and $\theta_{\text{bbf}}$ be the parameters of the argument extractor and black-box estimator modules. Then the gradient updates of the EstiNet during Online Training are:

$$\Delta_\theta = \frac{\partial L_{\text{target}}}{\partial \theta_{\text{arg}}} + \frac{\partial L_{\text{bbf}}}{\partial \theta_{\text{bbf}}}$$

Table 1: The Text-Logic task results: accuracy on the test data using baseline model and EstiNet with online training, at inference mode, both on varying amounts of training data.

| Train set size | 250 | 500 | 1,000 | 5,000 | 10,000 |
|---|---|---|---|---|---|
| **Baseline** | 0.533 | 0.686 | 0.859 | 0.931 | 0.98 |
| **EstiNet** | 0.966 | 0.974 | 0.968 | 0.995 | 1.0 |
| **Difference** | 81% | 41% | 13% | 7% | 2% |

To overcome this phenomenon, we follow Szegedy et al. (2016) and Pereyra et al. (2017), regularizing the high confidence by introducing (i) *Entropy Loss* – adding the negative entropy of the output distribution to the loss, therefore maximizing the entropy and encouraging less confident distributions, and (ii) *Label Smoothing Regularization* – adding the cross entropy (CE) loss between the output and the training set's label distribution (for example, uniform distribution) to the standard CE loss between the predicted and ground truth distributions. Empirical validation of the phenomenon and our proposed solution are detailed in Section 4.3.

## 4 EXPERIMENTS

We present four experiments in increasing complexity to test the Estimate and Replace approach and compare its performance against existing solutions. Specifically, the experiments demonstrate that by leveraging external black-box functions, we achieve better generalization and better learning efficiency in comparison with existing competing solutions, without using intermediate labels. Appendix A contains concrete details of the experiments.

### 4.1 TEXT-LOGIC

We start with a simple experiment that presents the ability of our Estimate and Replace approach to learn a proposed decomposition solution. We show that by leveraging a precise external function, our method performs better with less training data. In this experiment, we train a network to answer simple greater-than/less-than logical questions on real numbers, such as: "is 7.5 greater than 8.2?" We solve the text-logic task by constructing an EstiNet model with an argument extractor layer that extracts the arguments and operator (7.5, 8.2 and ">" in the above example), and a black-box estimator that performs simple logic operations (greater than and less than). We generate the Text-Logic questions from ten different templates, all requiring a true/false answer for two float numbers.

**Results** We compare the performance of the EstiNet model with a baseline model. This baseline model is equivalent to our model in its architecture, but is trained end-to-end with the task labels as supervision. This supervision allows the model to learn the input-to-output mapping, but does not provide any guidance for decomposing the task and learning the black-box function interface. We used online training for the EstiNet model. Table 1 summarizes the performance differences. The EstiNet model generalizes better than the baseline, and the accuracy difference between the two training procedures increases as the amount of training data decreases. This experiment presents the advantage of the Estimate and Replace approach to train a DNN with less data. For example, to achieve accuracy of 0.97, our model requires only 5% of the data that the baseline training requires.

### 4.2 IMAGE-ADDITION

With the second experiment we seek to present the ability of our Estimate and Replace approach to generalize by leveraging a precise external function. In addition, we compare our approach to an Actor Critic-based RL algorithm. The Image-Addition task is to sum the values captured by a sequence of MNIST images. Previously, Trask et al. (2018) have shown that their proposed Neural Arithmetic Logic Unit (NALU) cell generalizes better than previous solutions while solving this task[2] with standard end-to-end training. We solve the task by constructing an EstiNet model with an argument extractor layer that classifies the digit in a given MNIST image, and a black-box estimator that performs the sum operation. The argument extractor takes an unbounded series of MNIST

---

[2]They refer to this task as the MNIST-Addition task in their work.

Table 2: Loss performance for the image-addition task on the MNIST test-set. $k$ is the sequence length of the test set. Loss is mean absolute error. EstiNet results are in Inference mode.

| Model | k = 10 | k = 100 |
|---|---|---|
| NALU | 1.42 | 7.88 |
| EstiNet | 0.42 | 3.3 |

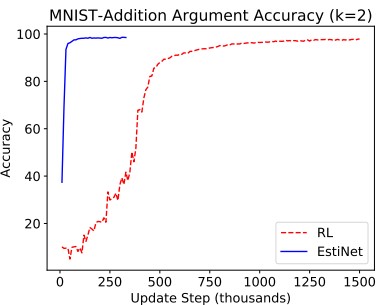 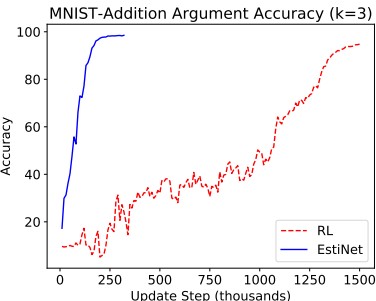

Figure 2: Learning efficiency of an Actor-Critic agent vs. the EstiNet model on the image-addition ($k \in \{2, 3\}$) task. Results show the MNIST test set classification accuracy of the argument extractor for RL Policy and for EstiNet as a function of the amount of gradient updates.

images as input, and outputs a sequence of MNIST classifications of the same length. The black-box estimator, which is a composition of a Long Short-Term Memory (LSTM) layer and a NALU cell, then takes the argument extractor's output as its input and outputs a single regression number. Solving the Image-Addition task requires the argument extractor to classify every MNIST image correctly without intermediate digit labels. Furthermore, because the sequence length is unbounded, unseen sequence lengths result in unseen sum ranges which the solution must generalize to.

**Results vs. End-to-End** Table 2 shows a comparison of EstiNet performance with an end-to-end NALU model. Both models were trained on sequences of length $k = 10$. The argument extractor achieves 98.6% accuracy on MNIST test set classification. This high accuracy indicates that the EstiNet is able to learn the desired $h^{\text{arg}}$ behavior, where the arguments are the digits shown in the MNIST images. Thus, it can generalize to any sequence length by leveraging the sum operation. Our NALU-based EstiNet outperforms the plain NALU-based end-to-end network.

**Results vs. RL** We compare the EstiNet performance with an AC-based RL agent as an existing solution for training a neural network calling a black-box function without intermediate labels. We compare the learning efficiency of the two models by the amount of gradient updates required to reach optima. Results in Figure 2 show that EstiNet significantly outperforms the RL agent.

### 4.3 IMAGE-LOOKUP

The third experiment tests the capacity of our approach to deal with non-differentiable tasks, in our case a lookup operation, as oppose to the differentiable addition operation presented in the previous section. With this experiment, we present the effect of replacing the black-box estimator with the original black-box function. We are given a $k$ dimensional lookup table $T : D^k \rightarrow D$ where $D$ is the digits domain in the range of $[0, 9]$. The image-lookup input is a sequence of length $k$ of MNIST images $(x_1, ..., x_k)$ with corresponding digits $(a_1, \ldots, a_k) \in D^k$. The label $y \in D$ for $(x_1, ..., x_k)$ is $T(a_1, \ldots, a_k)$. We solve the image-lookup task by constructing an EstiNet model with an argument extractor similar to the previous task and a black-box estimator that outputs the classification prediction.

**Results** Results are shown in Table 3. Successfully solving this task infers the ability to generalize to the black-box function, which in our case is the ability to replace or update the original lookup table with another at inference time without the need to retrain our model. To verify this we replace

Table 3: Accuracy results for the Image-Lookup task on the MNIST test-set for the three model configurations: train, test, and inference. We also report the accuracy of the argument extractor and estimator. The estimator accuracy is evaluated on the online sampling dataset. $k$ is the digit of MNIST images in the input, and the dimension of the lookup table.

| #MNIST images | Train | Test | Inference | Argument Extractor | Estimator |
|---|---|---|---|---|---|
| $k = 2$ | 0.98 | 0.11 | 0.97 | 0.99 | 0.98 |
| $k = 3$ | 0.97 | 0.1 | 0.97 | 0.99 | 0.98 |
| $k = 4$ | 0.69 | 0.1 | 0.95 | 0.986 | 0.7 |

the lookup table with a randomly generated one at test mode and observe performance decrease, as the black-box estimator did not learn the correct lookup functionality. However, in inference mode, where we replace the black-box estimator with the unseen black-box function, performance remains high. We also used the Image-Lookup task to validate the need for confidence regularization as described in Section 3.1.3. Figure 3 shows empirical results of correlation between over-confidence at the black-box estimator output distribution and small gradients corresponding to the argument extractor, as well as the vice versa when confidence regularizers are applied.

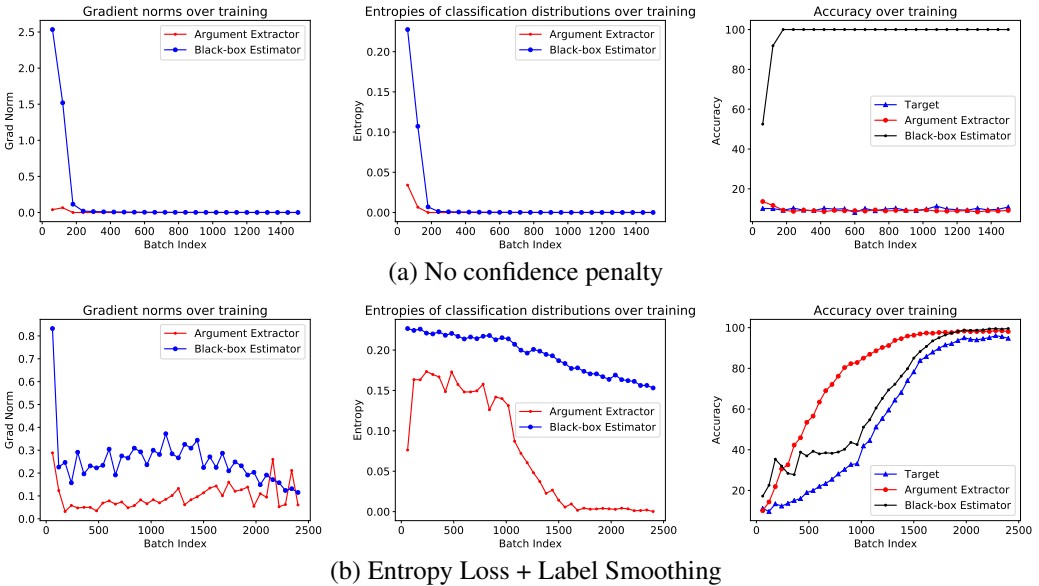

Figure 3: Empirical results for the confidence penalty in the image-lookup experiment for $k = 3$ (Section 4.3). The gradients norms, entropy and accuracy are shown as function of training time for identical models, with and without confidence regularization. The results show correlation between high confidence, measured as low entropies, and small gradients.

## 4.4 TEXT-LOOKUP-LOGIC (TLL)

For the last experiment, we applied the Estimate and Replace approach to solve a more challenging task. The task combines logic and lookup operations. In this task, we demonstrate the generalization ability on the input – a database table in this instance. The table can be replaced with a different one at inference time, like the black-box function from the previous tasks. In addition, with this experiment we compare the offline, online and hybrid training modes. For this task, we generated a table-based question answering dataset. For example, consider a table that describes the number of medals won by each country during the last Olympics, and a query such as: "Which countries won more than 7 gold medals?". We solve this task by constructing an EstiNet model with an argument extractor layer that (i) extracts the argument from the text, (ii) chooses the logical operation to perform (out

Table 4: Accuracy results for the text-lookup-logic task of the three model configurations: train, test, and inference with the training procedures: offline, online, and hybrid. Each value in the table is calculated as an average of ten repeated experiments.

| Training Type | Train | Test | Infer |
|---------------|-------|------|-------|
| **Offline**   | 0.09  | 0.02 | 0.17  |
| **Online**    | 0.76  | 0.22 | 0.69  |
| **Hybrid**    | 0.98  | 0.47 | 0.98  |

of: equal-to, less-than, greater-than, max and min), and (iii) chooses the relevant column to perform the operation on, along with a black-box estimator that performs the logic operation on the relevant column.

**Results**   Table 4 summarizes the TLL model performance for the training procedures described in Section 3.1. In offline training the model fails to fit the training set. Consequently, low training model accuracy results in low inference performance. We hypothesize that fixing the estimator parameters during the end-to-end training process prevents the rest of the model from fitting the train set. The online training procedure indeed led to significant improvement in inference performance. Hybrid training further improved upon online training fitting the training set and performance carried similarly to inference mode.

# 5   RELATED WORK

**End-to-End Learning**   Task-specific architectures for end-to-end deep learning require large datasets and work very well when such data is available, as in the case of neural machine translation (Bahdanau et al., 2014). General purpose end-to-end architectures, suitable for multiple tasks, include the Neural Turing Machine (Graves et al., 2014) and its successor, the Differential Neural Computer (Graves et al., 2016). Other architectures, such as the Neural Programmer architecture (Neelakantan et al., 2016) allow end-to-end training while constraining parts of the network to execute predefined operations by re-implementing specific operations as static differentiable components. This approach has two drawbacks: it requires re-implementation of the black-box function in a differentiable way, which may be difficult, and it lacks the accuracy and possibly also scalability of an exisiting black-box function. Similarly Trask et al. (2018) present a Neural Arithmetic Logic Unit (NALU) which uses gated base functions to allow better generalization to arithmetic functionality.

**Program Induction and Program Generation**   Program induction is a different approach to interaction with black-box functions. The goal is to construct a program comprising a series of operations based on the input, and then execute the program to get the results. When the input is a natural language query, it is possible to use semantic parsing to transform the query into a logical form that describes the program (Liang, 2016). Early works required natural language query-program pairs to learn the mapping, i.e., intermediate labels. Recent works, (e.g., Pasupat & Liang (2015)) require only query-answer pairs for training. Other approaches include neural network-based program induction (Andreas et al., 2016) translation of a query into a program using sequence-to-sequence deep learning methods (Lin et al., 2017), and learning the program from execution traces (Reed & De Freitas, 2015; Cai et al., 2017).

**Reinforcement Learning**   Learning to execute the right operation can be viewed as a reinforcement learning problem. For a given input, the agent must select an action (input to black-box function) from a set of available actions. The action selection repeats following feedback based on the previous action selection. Earlier works that took this approach include Branavan et al. (2009), and Artzi & Zettlemoyer (2013). Recently, Zaremba & Sutskever (2015) proposed a reinforcement extension to NTMs. Andreas et al. (2016) overcome the difficulty of discrete selections, necessary for interfacing with an external function, by substituting the gradient with an estimate using RL. Recent work by Liang et al. (2018) and Johnson et al. (2017) has shown to achieve state-of-the-art results in Semantic Parsing and Question Answering, respectively, using RL.

## 6    Discussion

**Interpretability via Composability**    Lipton (2016) identifies *composability* as a strong contributor to model interpretability. They define composability as the ability to divide the model into components and interpret them individually to construct an *explanation* from which a human can predict the model's output. The Estimate and Replace approach solves the black-box interface learning problem in a way that is modular by design. As such, it provides an immediate interpretability benefit. Training a model to comply with a well-defined and well-known interface inherently supports model composability and, thus, directly contributes to its interpretability.

For example, suppose you want to let a natural language processing model interface with a *WordNet* service to receive additional synonym and antonym features for selected input words. Because the WordNet interface is interpretable, the intermediate output of the model to the WordNet service (the words for which the model requested additional features) can serve as an explanation to the model's final prediction. Knowing which words the model chose to obtain additional features for gives insight to how it made its final decision.

**Reusability via Composability**    An additional clear benefit of model composability in the context of our solution is reusability. Training a model to comply with a well-defined interface induces well-defined module functionality which is a necessary condition for module reuse.

## 7    Conclusion

Current solutions for learning using black-box functionality in neural network prediction have critical limitations which manifest themselves in at least one of the following aspects: (i) poor generalization, (ii) low learning efficiency, (iii) under-utilization of available optimal functions, and (iv) the need for intermediate labels. In this work, we proposed an architecture, termed EstiNet, and a training and deployment process, termed Estimate and Replace, which aim to overcome these limitations. We then showed empirical results that validate our approach.

Estimate and Replace is a two-step training and deployment approach by which we first **estimate** a given black-box functionality to allow end-to-end training via back-propagation, and then **replace** the estimator with its concrete black-box function at inference time. By using a differentiable estimation module, we can train an end-to-end neural network model using gradient-based optimization. We use labels that we generate from the black-box function during the optimization process to compensate for the lack of intermediate labels. We show that our training process is more stable and has lower sample complexity compared to policy gradient methods. By leveraging the concrete black-box function at inference time, our model generalizes better than end-to-end neural network models. We validate the advantages of our approach with a series of simple experiments. Our approach implies a modular neural network that enjoys added interpretability and reusability benefits.

**Future Work**    We limit the scope of this work to tasks that can be solved with a single black-box function. Solving the general case of this problem requires learning of multiple black-box interfaces, along unbounded successive calls, where the final prediction is a computed function over the output of these calls. This introduces several difficult challenges. For example, computing the final prediction over a set of black-box functions, rather than a direct prediction of a single one, requires an additional network output module. The input of this module must be compatible with the output of the previous layer, be it an estimation function at training time, or its black-box function counterpart at inference time, which belong to different distributions. We reserve this area of research for future work.

As difficult as it is, we believe that artificial intelligence does not lie in mere knowledge, nor in learning from endless data samples. Rather, much of it is in the ability to extract the right piece of information from the right knowledge source for the right purpose. Thus, training a neural network to intelligibly interact with black-box functions is a leap forward toward stronger AI.

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

## APPENDIX A   EXPERIMENT DETAILS AND HYPERPARAMETERS

### A.1   IMAGE EXPERIMENTS

The Image-Addition and Image-Lookup tasks use the MNIST training and test sets. The input is a sequence of MNIST images, sampled uniformly from the training set. The black-box function is a sum operation which receives a sequence of digits in range $[0-9]$ represented as one-hot vectors. For Image-Lookup, the input sequence length defines the task (we've tested $k \in \{2, 3, 4\}$. $k = 4$ implies a lookup table of size $10^4$). For Image-Addition, we've trained on input length $k = 10$ and tested on $k = 100$. The implementation was done in PyTorch.

**Architecture**   The argument extractor for both tasks is a composition of two convolutional layers (conv$_1$, conv$_2$), each followed by local $2 \times 2$ max-pooling, and a fully-connected layer (fc$_{\text{arg}}$), which outputs the MNIST classification for an image. The argument extractors for each image share their parameters and each one outputs an MNIST classification for one image in the sequence. The sum estimator is an LSTM network, followed by a NALU cell on the final LSTM output, which results in a regression floating number. The lookup estimator is a composition of fully-connected layers (fc$_{\text{est}}^{\text{lookup}}$) with ReLU activations. The architecture parameters are detailed in Table 5.

| Argument Extractor | |
|---|---|
| conv$_1$ # filters | 10 |
| conv$_1$ filter size | $5 \times 5$ |
| conv$_1$ stride | 1 |
| conv$_2$ # filters | 20 |
| conv$_2$ filter size | $5 \times 5$ |
| conv$_2$ stride | 1 |
| fc$_{\text{arg}}$ dimensions | $[320, 10]$ |
| Lookup Estimator | |
| fc$_{\text{est}}^{\text{lookup}}$ dimensions | $[10k, 300, 100, 10]$ |
| Sum Estimator | |
| LSTM # layers | 1 |
| LSTM hidden size | 50 |
| NALU # layers | 1 |
| NALU hidden size | 100 |

Table 5: Image tasks architecture dimensions.

**Training**   We used the hybrid training procedure where the pre-training of the estimator (offline training) continued until either performance reached 90%, or stopped increasing, on synthetic 10-class (MNIST) distributions which were sampled uniformly. The hyper-parameters of the model are in Table 6. We note that confidence regularization was necessary to stabilize learning and mitigate vanishing gradients. The target losses are cross-entropy and squared distance for lookup and addition respectively. The loss functions are:

$$L_{\text{lookup}} = \text{LSR}(p, q; \epsilon) + \beta \text{LSR}(p_{\text{est}}, q_{\text{bbf}}; \epsilon) + \lambda \max(\sum_{\text{arg}} H(p_{\text{arg}}) - \Gamma, 0)$$

$$L_{\text{sum}} = (y' - y)^2 + \beta(y'_{\text{est}} - y_{\text{bbf}})^2 + \lambda \max(\sum_{\text{arg}} H(p_{\text{arg}}) - \Gamma, 0)$$

Where LSR stands for Label Smoothing Regularization loss, $H$ stands for entropy, $p$ stands for the output classification, $q$ stands for the gold label (one-hot), and $y'$ and $y$ stand for the model and gold MNIST sum regressions, respectively. The $\beta$-weighted component of the loss is the online loss. The $\lambda$-weighted component is threshold entropy loss regularization on the argument extractor's MNIST classifications.

In the following we describe the RL environment and architecture used in our experiments. We employed fixed length episodes and experimented with $k \in \{2, 3\}$. The MDP was modeled as follows: at each step a sample $(x_t, y_t)$ is randomly selected from the MNIST dataset, where the

| Parameter | Addition | Lookup |
|---|---|---|
| Online loss | $\beta = 1.0$ | $\beta = 1.0$ |
| Entropy loss | $\lambda = 0.15$ | $\lambda = 0.1$ |
| Entropy loss threshold | $\Gamma = 0.15$ | $\Gamma = 0.15$ |
| LSR confidence penalty | — | $\epsilon = 0.6$ |
| LSR label distribution prior | — | Uniform |
| Optimizer | Adam | Adam |
| Learning rate | 0.001 | 0.001 |
| Batch size | 50 | 20 |

Table 6: Image tasks hyperparameters.

handwritten image is used as the state, i.e. $s_t = x_t$. The agent responds with an action $a_t$ from the set $[0, 9]$. The reward, $r_t$, in all steps except the last step is 0, and equals to the sum of absolute errors between the labels of the presented examples and the agent responses in the last step:

$$r_k = \| \sum_{t=1}^{k} (a_t - y_t) \|$$

Where $y_t$ is the digit label.

We use A3C as detailed by Mnih et al. (2016) as the learning algorithm containing a single worker which updates the master network at the end of each episode. The agent model was implemented using two convolutional layers with filters of size $5 \times 5$ followed by a max-pooling size $2 \times 2$. The first convolutional layer contains 10 filters while the second contains 20 filters. The last two layers were fully connected of sizes 256 and 10 respectively with ELU activation, followed by a softmax. We employed Adam optimization (Kingma & Ba, 2014) with learning rate $1e - 5$.

## A.2 TEXT EXPERIMENTS

The Text-Logic and Text-Lookup-Logic experiments were implemented in TensorFlow on synthetic datasets generated from textual templates and sampled numbers. We give concrete details for both experiments.

### A.2.1 TEXT-LOOKUP-LOGIC

For the TLL task we generated a table-based question answering dataset. The TLL dataset input has two parts: a question and a table. To correctly answer a question from this dataset, the DNN has to access the right table column and apply non-differentiable logic on it using a parameter it extracts from the query. For example, consider a table that describes the number of medals won by each country during the last Olympics, and a query such as: "Which countries won more than 7 gold medals?" To answer this query the DNN has to extract the argument (7 in this case) from the query, access the relevant column (namely, gold medals), and execute the greater than operation with the extracted argument and column content (namely a vector of numbers) as its parameters. The operation's output vector holds the indexes of the rows that satisfy the logic condition (greater-than in our example). The final answer contains the names of the countries (i.e., from the countries column) in the selected rows.

**The black-box function interface** Solving the TLL task requires five basic logic functions: equal-to, less-than, greater-than, max, and min. Each such function defines an API that is composed of two inputs and one output. The first input is a vector of numbers, namely, a column in the table. The second is a scalar, namely, an argument from the question or NaN if the scalar parameter is not relevant. The output is one binary vector, the same size as the input vector. The output vector indicates the selected rows for a specific query and thus provides the answer.

**TLL data** We generated tables in which the first row contains column names and the first column contains a list of entities (e.g., countries, teams, products, etc.). Subsequent columns contained the quantitative properties of an entity (e.g., population, number of wins, prices, discounts, etc.). Each TLL-generated table consisted of 5 columns and 25 rows. We generated entity names (i.e., nations and clubs) for the first column by randomly selecting from a closed list. We generated values for the

rest of the columns by sampling from a uniform distribution. We sampled values between 1 and 100 for the train set tables, and between 300 and 400 for the test set tables. We further created 2 sets of randomly generated questions that use the 5 functions. The set includes 20,000 train questions on the train tables and 4,000 test questions on the test tables.

**Input representations** The TLL input was composed of words, numbers, queries, and tables. We used word pieces as detailed by Wu et al. (2016) to represent words. A word is a concatenation of word pieces: $w_j \in \mathbb{R}^d$ is an average value of its piece embedding.

The exact numerical value of numbers is important to the decision. To accurately represent a number and embed it into the same word vector space, we used number representation following the float32 scheme (Kahan, 1996). Specifically, it starts by representing a number $a \in \mathbb{R}$ as a 32 dimension Boolean vector $s'_n$. It then adds redundancy factor $r, r * 32 < d$ by multiplying each of the $s'_n$ digits $r$ times. Last, it pads the $s_n \in \mathbb{R}^d$ resulting vector with $d - r * 32$ zeros. We tried several representation schemes. This approach resulted in the best EstiNet performance.

We represent the query $q$ as a matrix of word embeddings and use an LSTM model (Hochreiter & Schmidhuber, 1997) to encode the query matrix into a vector representation: $q_{\text{lstm}} \in \mathbb{R}^{d_{\text{rnn}}} = h_{\text{last}}(\text{LSTM}(Q))$ where $h_{\text{last}}$ is the last LSTM output and $d_{\text{rnn}}$ is the dimension of the LSTM.

Each table $T \in \mathbb{R}^{n \times m \times d}$ with $n$ rows and $m$ columns is represented as a three dimensional tensor. It represents a cell in a table as the piece average of its words.

**Argument Extractors Architecture** The EstiNet TLL model uses three types of "selectors" (argument extractors): operation, argument, and column. Operation selectors select the correct black-box function. Argument selectors select an argument from the query and hand it to the API. The column selector's role is to select a column from the table and pass it to the black-box function. We implement each selector subnetwork as a classifier. Let $C \in \mathbb{R}^{c_n \times d_c}$ be the predicted class matrix, where the total number of classes is $c_n$ and each class is represented by a vector of size $d_c$. For example, for a selector that has to select a word from a sentence, the $C$ matrix contains the word embeddings of the words in the sentence. One may consider various selector implementation options. We use a simple, fully connected network implementation in which $W \in \mathbb{R}^{d_{\text{rnn}} \times c_n}$ is the parameter matrix and $b \in \mathbb{R}^{d_c}$ is the bias. We define $\beta = C(q_{\text{lstm}}W + b)$ to be the selector prediction before activation and $\alpha = f_{\text{sel}}(.) = \text{softmax}(\beta)$ to be the prediction after the softmax activation layer. At inference time, the selector transforms its soft selection into a hard selection to satisfy the API requirements. EstiNet enables that using Gumbel Softmax hard selection functionality.

**Estimator Architecture** We use five estimators to estimate each of the five logic operations. Each estimator is a general purpose subnetwork that we implement with a transformer network encoder Vaswani et al. (2017). Specifically, we use $n \in \mathbb{N}$ identical layers, each of which consists of two sub-layers. The first is a multi-head attention with $k \in \mathbb{N}$ heads, and the second is a fully connected feed forward two-layer network, activated separately on each cell in the sequence. We then employ a residual connection around each of these two sub-layers, followed by layer normalization. Last, we apply linear transformation on the encoder output, adding bias and applying a Gumbel Softmax.

## A.2.2 TEXT-LOGIC

The task input is a sentence that contains a greater-than or less-than question generated from a set of ten possible natural language patterns. The argument extractor must choose the correct tokens from the input to pass to the estimator/black-box function, which executes the greater-than/less-than functionality. For example: *Out of x and y , is the first bigger ?* where $x, y$ are float numbers sampled from a $\sim \mathcal{N}(0, 10^{10})$ distribution. The architecture is a very simple derivative of the TLL model with two selectors for the two floating numbers, and a classification of the choice between greater-than or less-than.

