# OpenReview forum: "Neural network gradient-based learning of black-box function interfaces"
_ICLR.cc/2019/Conference_

### Official Review · AnonReviewer2 · 2018-10-31
**A good idea with proper validation**

**Rating:** 7
**Confidence:** 3

**Review:**

The paper proposes a method to solve end-to-end learning tasks using a combination of deep networks and domain specific black-box functions. In many machine learning tasks there may be a sub-part of the task can be easily solved with a black-box function (e.g a hard coded logic).  The paper proposes to use this knowledge in order to design a deep net that mimics the black-box function. This deep net being differentiable can be utilized while training in order to perform back-propagation for the deep nets that are employed to solve the remaining parts of the task.

The paper is well written and in my opinion the experiments are solid. They show significant gains over well-designed baselines. (It should be noted that I am not super familiar with prior work in this area and may not be aware of some related baselines that can be compared with.)

In Section 3.1.2 the authors discuss offline and online methods to train the mimicking deep network of a black-box function. The offline version suffers from wasting samples on unwanted regions while the online version will have a cold-start problem. However, I believe there can be better solution than the hybrid strategy. In fact there is a clear explore/exploit trade-off  here. Therefore, one may start with a prior over the input domain of the black-box function and then as the argument extractor learns well the posterior can be updated. Then we can Thompson sample the inputs from this posterior in order to train the mimicking network.  I think such a bandit inspired approach will be interesting to try out.

---

> ### Author Response · Authors · 2018-11-25
> **Reply to the reviewer's suggestion**
>
> Thank you for your detailed and positive review.
>
> Regarding your last comment, if we have understood it correctly, you are suggesting to use the argument extractor when sampling to the estimator during training for better exploration of the argument space. We believe that this has potential to improve sample complexity. The hybrid training has served us well enough in our experiments, but it is a great idea for future work. One challenge is that it is not immediately clear what the initial prior should be (and how strong when updating the posterior).
>
> Thank you again for your comments
> - Authors

---

### Official Review · AnonReviewer3 · 2018-11-04
**Well written paper and convincing results**

**Rating:** 7
**Confidence:** 3

**Review:**

This paper is about training a neural network (NN) to perform regression given a dataset (x, y) *and* a black box function which we know correctly maps from some intermediate representation to y. Instead of learning a NN directly from x to y, we want to make use of this black box function and learn a mapping from x to the intermediate representation. Call this the "argument extractor" NN. The problem is that (i) the black box function is typically non-differentiable so we cannot learn end-to-end and (ii) we don't have labels for the intermediate representations in order to learn a NN to approximate the black box function. The authors propose to train in three different ways: (1) offline training: train an auxiliary NN that approximates the black box function based on data generated by sampling the input uniformly (or similar); then train both the auxiliary NN and the argument extractor NN together end-to-end using (x, y) data, (2)  online training: train the auxiliary NN and the argument extractor NN together, based on (x, y) data; data for training the auxiliary NN comes from the argument extractor NN during the main training, and (3) hybrid training: pre-train the auxiliary NN as in (1) and then train both NNs as in (2).

Experimental results show:
- this approach leads to better performance than regressing directly from x to y in the small data regime,
- this approach leads to better generalization (being able to add more image numbers during test),
- this approach learns faster than an actor-critic based RL agent,
- this approach can be useful even if the functionality of the black-box function inherently cannot be estimated by a differentiable function (lookup table) - the resulting argument extractor NN is useful when used with the non-differentiable black box function,
- hybrid training is the best; offline training is the worst,
- penalizing low output entropy helps.

It wasn't quite clear to me which training procedure was used for experiments 4.1-4.3. Presumably hybrid? It would also be nice to see how much time is spent in pre-training vs main training. In figure 2, what are the update steps for EstiNet (since there are two losses + pretraining)?

I found this paper to be generally well-written and results convincing.

---

> ### Author Response · Authors · 2018-11-25
> **Appendix added to clarify experiment details**
>
> Thank you for the detailed and positive review.
>
> To answer your questions regarding specific experiment details, we have added an appendix which contains all of the relevant details. Specifically, we have indeed used hybrid training for experiments 4.2, 4.3, where the pre-training continued until either a satisfactory performance was reached, such as 90%, or performance stopped increasing (the behavior depends on the difficulty of the task and quality of the offline sampling). Experiment 4.1 uses online training (this was specified in Table 1's caption but we have now added the information to the main text).
>
> The pre-training took a negligible amount of time in comparison to the target/online training since it relied on a sub-component of the network and loss (additionally, the arguments domain is far smaller than the input domain).
>
> The exact loss function is also now detailed in the appendix, but in essence, we used a direct sum (i.e. weights of 1.0) of the target loss and online loss.
>
> Thank you again for your comments
> - Authors

---

### Official Review · AnonReviewer1 · 2018-11-06
**Interesting approach with good results on synthetic tasks**

**Rating:** 7
**Confidence:** 3

**Review:**

This paper presents an approach, called EstiNet, to train a hybrid models which uses both neural networks and black-box functions. The key idea is that, during training, a neural network can be used to approximate the functionality of the black-box functions, which makes the whole system end-to-end differentiable. At test time, the true black-box functions are still used. The training objective composes two parts: L_bbf, the loss for approximating the black-box function and L_target, the loss for the end-to-end goal. They tried different variations of when to train the black-box function approximator or not. It is shown to outperform the baselines like end-to-end differentiable model or NALU over 4 synthetic tasks in sample efficiency. There are some analysis about how the entropy loss and label smoothing helps with the gradient flow.

The proposed model is interesting, and is shown to be effective in the synthetic tasks. The paper is well-written and easy to follow. However, some of the experiment details are missing or scattered in the text, which might make it hard for the readers to reproduce the result. I think it helps to have the experimental details (number of examples, number of offline pretraining steps, size of the neural network, etc) organized in some tables (could be put in the appendix).

Two main concerns about how generally applicable is the proposed approach:

1. It helps to show how L_target depends on L_bbf, or how good the approximation of the black-box function has to be to make the approach applicable. For example, some functions, such as sorting, are hard to approximate by neural network in a generalizable way, so in those cases, is it still possible to apply the proposed approach?

2.The proposed approach can be better justified by discussing some potential real world applications. Two closely related applications I can think of are visual question answering and semantic parsing. However, it is hard to find good black-box functions for VQA and people often learn them as neural networks, and the functions in semantic parsing often need to interact with a database or knowledge graph, which is hard to approximate with a neural network.

Some minor issues:

Table 3 isn’t very informative since k=2 and k=3 provides very similar results. It would help to show how large k needs to be for the performance to severely degrade.

Missing references:

The Related Works section only reviewed some reinforcement learning work on synthetic tasks. However, with some bootstrapping, RL is also shown to achieve state-of-the-art performance on visual question answering and semantic parsing tasks (Johnson et al, 2017; Liang et al, 2018), which might be good to include here.

Johnson, J., Hariharan, B., van der Maaten, L., Hoffman, J., Fei-Fei, L., Zitnick, C. L., & Girshick, R. B. (2017, May). Inferring and Executing Programs for Visual Reasoning. In ICCV (pp. 3008-3017).
Liang, C., Norouzi, M., Berant, J., Le, Q., & Lao, N. (2018). Memory augmented policy optimization for program synthesis with generalization. arXiv preprint arXiv:1807.02322.

---

> ### Author Response · Authors · 2018-11-25
> **Paper revised (including experiment appendix) to address comments**
>
> Thank you for the very detailed criticism and positive review.
>
> We have updated the paper to address your concerns in the following way:
>
> (1) We have added an appendix section with experimental details for the purpose of reproducing the results. While we cannot make a concrete promise, we will make a concerted effort to release the code.
>
> (2) We refer to both of your points (1) and "minor" together:
>
> We have added more detail to Table 3 in the form of a k=4 experiment and an additional column for the estimator accuracy on the online sampling dataset. We could not execute a k=5 experiment because of resource constraints in training an estimator network for a 10^5 lookup table. The experiment results are:
> Image-Lookup k=4
> Train: 0.69
> Test: 0.1
> Inference: 0.95
> Argument extractor: 0.986
> Estimator: 0.7
>
> In this experiment, the bbf estimator only reaches a performance of 0.7 (in other words, it learns about 70% of the values in the 10^4 lookup table), which proves enough for the argument extractor to learn the desired functionality, allowing the network to perform well at inference time with the real black-box function. This result should help address your concern in point (1). We note that it is not necessary for the estimator to learn in a way that generalizes to unseen inputs (because it is discarded after training), as long as it performs the correct mapping on the training set.
>
> It is true that learning is dependent on the performance of the estimator. Whether the argument extractor can learn from an imperfect estimator is likely dependent on the ratio of noisy signals (from incorrect estimator decisions) to informative signals, and the ability of the interface between them to generalize from the informative examples to the noisy examples. In the case where the estimator is never correct for any decision of one of the argument extractors, that specific argument extractor will be unable to learn (in the Image-Lookup case, this would mean the estimator is incorrect for an entire dimension slice of the lookup bbf tensor).
>
> (3) Regarding the practicality of the approach towards real-world tasks like Semantic Parsing and Question Answering, it has indeed been our main motivation for this work. The synthetic Text-Lookup-Logic experiment was meant to serve as a first step in that direction. We've added a short mention of these motivations in the introduction.
>
> We have also appended the Related Works section with your suggestions.
>
> The new revision of the paper has been uploaded to this page.
>
> Thank you again for your comments
> - Authors

---

> > ### Comment · AnonReviewer1 · 2018-11-26
> > **Response to authors' reply**
> >
> > I have read the authors' reply. I am generally happy with the revision and will keep my rating.

---

### Author Response · Authors · 2018-11-25
**Revision change-log**

We thank all of the reviewers and we apologize for the late response.

To address comments of the reviewers, the PDF file of the paper has been updated with the following changes:

- We have added an appendix that contains the implementation details of our experiments, including architecture and hyperparameters.
- Another experiment was added to Table 3 in Section 4. It is an additional k=4 experiment for the Image-Lookup task, and its primary addition is an example of an instance where the argument extractor was able to learn its desired functionality in spite of an imperfect estimator.
- Added citations in the Related Works section (under RL).

As well as some small typo fixes.

Thanks for all of the great comments

- Authors

---

### Meta-Review · Area_Chair1 · 2018-12-17
**Important problem setup and strong evaluation**

**Confidence:** 4
**Recommendation:** Accept (Poster)

**Metareview:**

The paper focuses on hybrid pipelines that contain black-boxes and neural networks, making it difficult to train the neural components due to non-differentiability. As a solution, this paper proposes to replace black-box functions with neural modules that approximate them during training, so that end-to-end training can be used, but at test time use the original black box modules. The authors propose a number of variations: offline, online, and hybrid of the two, to train the intermediate auxiliary networks. The proposed model is shown to be effective on a number of synthetic datasets.

The reviewers and AC note the following potential weaknesses: (1) the reviewers found some of the experiment details to be scattered, (2) It was unclear what happens if there is a mismatch between the auxiliary network and the black box function it is approximating, especially if the function is one, like sorting, that is difficult for neural models to approximate, and (3) the text lacked description of real-world tasks for which such a hybrid pipeline would be useful.

The authors provide comments and a revision to address these concerns. They added a section that described the experiment setup to aid reproducibility, and incorporated more details in the results and related work, as suggested by the reviewers. Although these changes go a long way, some of the concerns, especially regarding the mismatch between neural and black box function, still remain.

Overall, the reviewers agreed that the issues had been addressed to a sufficient degree, and the paper should be accepted.